# NovelD: A Simple yet Effective Exploration Criterion

**Tianjun Zhang**[1]    **Huazhe Xu**[1]    **Xiaolong Wang**[2]    **Yi Wu**[3,5]    **Kurt Keutzer**[1]

**Joseph E. Gonzalez**[1]              **Yuandong Tian**[4]

[1]University of California, Berkeley    [2]Unveristy of California, San Diego

[3]Tsinghua University    [4]Facebook AI Research    [5]Shanghai Qi Zhi Institue

## Abstract

Efficient exploration under sparse rewards remains a key challenge in deep reinforcement learning. Previous exploration methods (e.g., RND) have achieved strong results in multiple hard tasks. However, if there are multiple novel areas to explore, these methods often focus quickly on one without sufficiently trying others (like a depth-wise first search manner). In some scenarios (e.g., four corridor environment in Sec. 4.2), we observe they explore in one corridor for long and fail to cover all the states. On the other hand, in theoretical RL, with optimistic initialization and the inverse square root of visitation count as a bonus, it won't suffer from this and explores different novel regions alternatively (like a breadth-first search manner). In this paper, inspired by this, we propose a simple but effective criterion called NovelD by weighting every novel area approximately equally. Our algorithm is very simple but yet shows comparable performance or even outperforms multiple SOTA exploration methods in many hard exploration tasks. Specifically, NovelD solves all the static procedurally-generated tasks in Mini-Grid with just 120M environment steps, without any curriculum learning. In comparison, the previous SOTA only solves 50% of them. NovelD also achieves SOTA on multiple tasks in NetHack, a rogue-like game that contains more challenging procedurally-generated environments. In multiple Atari games (e.g., MonteZuma's Revenge, Venture, Gravitar), NovelD outperforms RND. We analyze NovelD thoroughly in Mini-Grid and found that empirically it helps the agent explore the environment more uniformly with a focus on exploring beyond the boundary. [1]

## 1 Introduction

Deep reinforcement learning (RL) has experienced significant progress over the last several years, with an impressive performance in games like Atari [41, 4], StarCraft [61], Go and Chess [57–59]. However, its success often requires massive computational resources or manually designed dense rewards. The dense rewards are often impractical for real-world settings as they require a significant amount of task-specific domain knowledge.

An alternative approach is to allow the agent to explore the environment freely until it reaches the goal. While basic RL exploration criteria (e.g., $\epsilon$-greedy) are quite simple, it fails to explore sufficiently in hard tasks. Modern works adopt various Intrinsic Reward (IR) designs to guide exploration in hard-exploration settings. However, we observe (Sec. 4.2) in our experiments if there are multiple

---

[1]Correspondence to: Tianjun Zhang <tianjunz@berkeley.edu>. Our code is available at `https://github.com/tianjunz/NovelD`.

35th Conference on Neural Information Processing Systems (NeurIPS 2021).

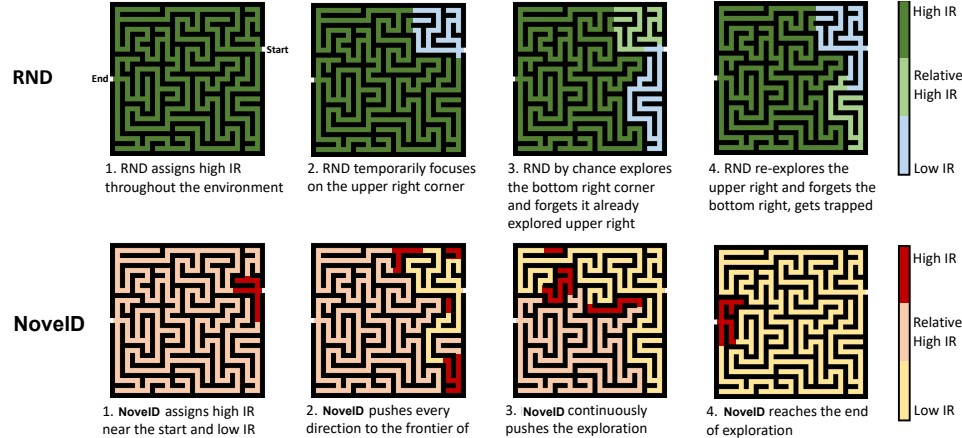

Figure 1: A hypothetical demonstration of how exploration is done in NovelD versus Random Network Distillation (RND) [11], in terms of distribution of intrinsic reward (IR). NovelD reaches the goal by continuously pushing the frontier of exploration while RND gets trapped in the explored regions. Note that IR is defined differently in RND versus NovelD (See Eqn. 2) and different colors are used to represent IR of RND and NovelD.

regions of interest, these methods sometimes quickly are trapped in one area without sufficiently exploring others. This results in poor overall exploration of large state space [2]. This is also known as *detachment* problem in Go-Explore [17].

On the other hand, provably optimal theoretical RL equipped with optimistic initialization and visitation count bonus explores the environment much more uniformly [31]. Inspired by that, we introduce a very simple but effective exploration criterion by weighting each novel area approximate equally. Our algorithm uses regulated **Novel**ty **D**ifference (**NovelD**) of consecutive states in a trajectory. The novelty of a state is calculated by Random Network Distillation (RND [11]). The underlying intuition is that this criterion provides a large intrinsic reward at the *boundary* between the explored and the unexplored regions (Fig. 1). As a result, it induces a very different exploration pattern comparing to count-based approaches [6, 10, 11, 48, 5] and yields a much broader coverage over the state space.

We evaluate **NovelD** on three very challenging exploration environments: MiniGrid [13], NetHack [34] and Atari games [9]. MiniGrid is a popular benchmark for evaluating exploration algorithms [52, 12, 24]; NetHack, built from a real game, is a much more realistic, PG-generated environment with complex goals and skills. Atari games is a widely used benchmark for RL algorithms [40, 11, 18]. NovelD manages to solve all the static environments in MiniGrid within 120M environment steps, without curriculum learning. In contrast, previous SOTA AMIGo [12] solves 50% of the tasks, categorized as "easy" and "medium", by training a separate goal-generating teacher network in 500M steps. In NetHack, NovelD also outperforms all baselines with a significant margin on various tasks. NovelD is also tested in various Atari games (e.g., MonteZuma's Revenge, Venture), using image-based input and outperforms RND for both CNN and RNN-based networks.

Compared to previous works (e.g., RIDE [52], AMIGo [12] and Go-Explore [17]), NovelD has a few design advantages: **(1)** in NovelD, there is almost no hyperparameters; **(2)** NovelD is one-stage approach and can be readily combined with any policy learning methods (e.g., PPO), while many approaches (e.g., RIDE, Go-Explore) are two-stage approaches. **(3)** NovelD is asymptotic consistent: its IR vanishes after sufficient exploration, while approaches like RIDE and AMIGo do not. During our experiments, we observe that compared to the count-based approach and RND, NovelD prioritizes the unexplored boundary states, yielding much more efficient and broader exploration patterns.

## 2 Background

In a Markov Decision Process (MDP), we define state space $S$, action space $A$, and (non-deterministic) transition function $\mathcal{T} : S \times A \rightarrow P(S)$ where $P(S)$ is the probability of next state given the current state and action. The goal is to maximize the expected reward $R = \mathbb{E}[\sum_{k=0}^{T} \gamma^k r_{t+k=1}]$ where $r_t$ is the reward and $\gamma$ is the discount factor. In this paper, the total reward received at time step $t$ is given by $r_t = r_t^e + \alpha r_t^i$, where $r_t^e$ is the extrinsic reward given by the environment, $r_t^i$ is the intrinsic reward from the exploration criterion, and $\alpha$ is a scaling hyperparameter.

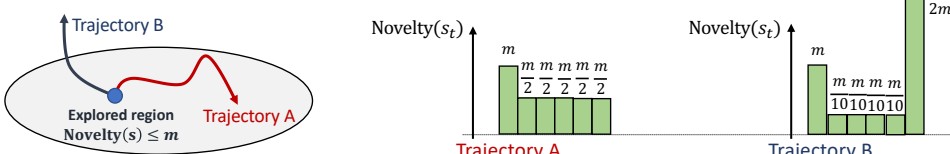

Figure 2: The mechanism of NovelD. Two trajectories, A and B. Trajectory A is within the "explored region" defined as the subset of states with $\text{novelty}(\mathbf{s}) \leq m$ for some constant $m$ and tends to have high accumulated novelty, while Trajectory B explores out of explored regions. RND assigns comparable IR on A $((1 + 5/2)m = 3.5m)$ and B $((1 + 4/10 + 2)m = 3.4m)$, while NovelD assigns much higher IR on B $((2 - 1/10)m = 1.9m)$ than A (0). Moreover, this effect becomes more prominent for longer trajectories. The value of $\alpha$ in Eq. 1 is a design choice and we use $\alpha = 1$ here for showing an intuitive example.

## 3 Exploration via Novelty Difference

### 3.1 NovelD Intrinsic Reward Criterion

NovelD is a meta-criterion that can be applied on top of any *novelty measure* $\text{novelty}(\mathbf{s})$ that tells how familiar a state $\mathbf{s}$ is for the agent. Formally, along a trajectory, we give agent IR if the previous state $\mathbf{s}_t$ has been sufficiently explored but $\mathbf{s}_{t+1}$ is not:

$$r^i(\mathbf{s}_t, \mathbf{a}_t, \mathbf{s}_{t+1}) = \max\left[\text{novelty}(\mathbf{s}_{t+1}) - \alpha \cdot \text{novelty}(\mathbf{s}_t), 0\right], \tag{1}$$

Here $\alpha$ is a scaling factor. Intuitively, if we define $\{\mathbf{s} : \text{novelty}(\mathbf{s}) \leq m\}$ to be an explored region, then NovelD gives intrinsic reward (IR) to the agent when it explores beyond the boundary of explored regions. Note that IR is clipped to avoid negative IR if the agent transits back from a novel state to a familiar state. From the equation, only crossing the frontier matters to the intrinsic reward; if both $\mathbf{s}_{t+1}$ and $\mathbf{s}_t$ are novel or familiar, their difference would be small. For each trajectory going towards the frontier/boundary, NovelD assigns an approximately equal IR, regardless of the length (see Sec. 3.2). Like RIDE [52], in our actual implementation, partial observation $\mathbf{o}_t$ are used instead of the true state $\mathbf{s}_t$, when $\mathbf{s}_t$ is not available.

**Episodic Restriction on Intrinsic Reward (ERIR).** Simply using Eqn. 1 to guide exploration can result in the agent going back and forth between novel states $\mathbf{s}_{t+1}$ and their previous states $\mathbf{s}_t$. RIDE [52] avoids this by scaling the intrinsic reward $r^i(\mathbf{s})$ by the inverse of the episodic state visitation counts. NovelD puts a more aggressive restriction: the agent is only rewarded when it visits the state $\mathbf{s}$ for the first time in an episode. Thus, the intrinsic reward of NovelD becomes:

$$r^i(\mathbf{s}_t, \mathbf{a}_t, \mathbf{s}_{t+1}) = \max\left[\text{novelty}(\mathbf{s}_{t+1}) - \alpha \cdot \text{novelty}(\mathbf{s}_t), 0\right] * \mathbb{I}\{N_e(\mathbf{s}_{t+1}) = 1\} \tag{2}$$

$N_e$ here stands for episodic state count and is reset every episode. In contrast, the novelty measure $\text{novelty}(\cdot)$ is a life-long counter throughout training.

**Novelty Measure using Random Network Distillation.** How to accurate measure the novelty of a state $\mathbf{s}$ in large-scale stochastic environments remains an open problem [6]. Similar to RND [11], we use the difference between a random fixed target network $\phi$ and a trainable predictor network $\phi'_{\boldsymbol{w}}$ as a novelty measure:

$$\text{novelty}(\mathbf{s}_t) = \text{novelty}(\mathbf{s}_t; \boldsymbol{w}) := \|\phi(\mathbf{s}_t) - \phi'_{\boldsymbol{w}}(\mathbf{s}_t)\|_2, \tag{3}$$

Once the novelty score for state $\mathbf{s}_t$ is computed, we minimize $\text{novelty}(\mathbf{s}_t; \boldsymbol{w})$ with respect to $\boldsymbol{w}$ and perform a one-step weight update for $\boldsymbol{w}$, which is the parameter of the predictor network $\phi'_{\boldsymbol{w}}$. Therefore, when $\mathbf{s}_t$ is visited again, the quantity $\text{novelty}(\mathbf{s}_t; \boldsymbol{w})$ will be lower, showing that the state has been seen in the past.

### 3.2 Conceptual Advantages of NovelD over Existing Criteria

We show the distinct preference of exploration directions between RND and NovelD. We also see that NovelD is a consistent algorithm, the intrinsic reward converges to zero after sufficient exploration.

**Explore via Novelty Difference.** Fig. 2 shows the conceptual comparison between NovelD (Eq. 2) and exploration with novelty measure alone (e.g., RND). We could clearly see their distinctive preferences: our criterion tends to reward more the trajectories that move out of the explored regions (Trajectory B), which typically show a profile of stable novelty measure until a sudden boost when it ventures into an unknown region. On the other hand, count-based approaches can be trapped within explored regions where the trajectories have low but consistent novelty rewards (Trajectory A).

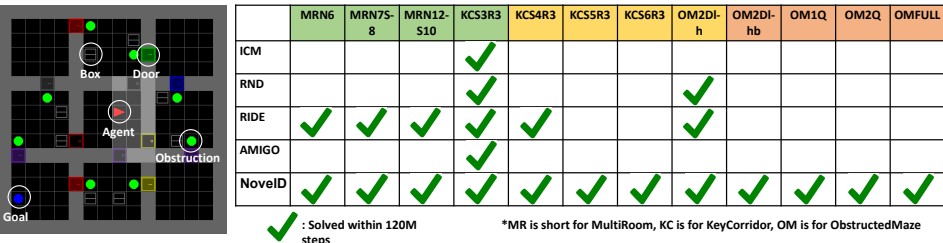

| | MRN6 | MRN7S-8 | MRN12-S10 | KCS3R3 | KCS4R3 | KCS5R3 | KCS6R3 | OM2Dl-h | OM2Dl-hb | OM1Q | OM2Q | OMFULL |
|---|---|---|---|---|---|---|---|---|---|---|---|---|
| ICM | | | | ✓ | | | | | | | | |
| RND | | | | ✓ | | | | ✓ | | | | |
| RIDE | ✓ | ✓ | ✓ | ✓ | ✓ | | | ✓ | | | | |
| AMIGO | | | | ✓ | | | | | | | | |
| NovelD | ✓ | ✓ | ✓ | ✓ | ✓ | ✓ | ✓ | ✓ | ✓ | ✓ | ✓ | ✓ |

✓ : Solved within 120M steps          *MR is short for MultiRoom, KC is for KeyCorridor, OM is for ObstructedMaze

Figure 3: MiniGrid Environments. **Left:** a procedurally-generated `OMFull` environment. **Right:** NovelD solves challenging tasks which previous approaches cannot solve. Note that we evaluate all methods for 120M steps. AMIGo gets better results when trained for 500M steps as shown in [12], but is still not as good as results obtained by NovelD in 120M steps.

**Asymptotic Inconsistency.** Approaches that define IR as the difference between state representations $\|\psi(\mathbf{s}_t) - \psi(\mathbf{s}_{t+1})\|$ ($\psi$ is a learned embedding network) [63, 38] suffer from asymptotic inconsistency. In other words, their IR does not vanish even after sufficient exploration: $r^i \not\to 0$ when $N \to \infty$. This is because when the embedding network $\psi$ converges after sufficient exploration, the agent can always obtain non-zero IR if a significant change occurs in state representation (e.g., opening a door or picking up a key in MiniGrid). Therefore, the learned policy does not maximize the extrinsic reward $r^e$, deviating from the goal of RL. Automatic curriculum approaches [12]) have similar issues due to an ever-present IR. In contrast, our Eq. 4 is asymptotically consistent as $r^i \to 0$ when $N \to \infty$.

## 4 Experiments

We evaluate NovelD on challenging procedurally-generated environment MiniGrid [13], several hard-exploration scenarios in Atari games, and NetHack [34] with sparse rewards (i.e., get reward only when reaching the final goal). For all the experiments and all exploration approaches (including NovelD), we use PPO [55] as the base RL algorithm and add intrinsic reward specified by various methods, to encourage exploration.

In MiniGrid, we compare NovelD with RND [11], ICM [49], RIDE [52] and AMIGo [12]. We only evaluate AMIGo for 120M steps in our experiments. The algorithm obtains better results when trained for 500M steps as shown in [12]. For all other baselines, we follow the same training paradigm from [52]. Mean and standard deviation across four runs of different seeds are computed.

NovelD solves all the static tasks provided by MiniGrid. In contrast, all the baselines end up with zero rewards on half of the tasks we tested. In NetHack, NovelD achieves SOTA with a large margin over baselines (IMPALA [19] without exploration bonus and RND).

### 4.1 MiniGrid Environments

We mainly use three challenging environments from MiniGird: *Multi-Room* (**MR**), *Key Corridor* (**KC**) and *Obstructed Maze* (**OM**). We use these abbreviations for the rest of the paper (e.g., `OM2Dlh` stands for ObstructedMaze2Dlh). Fig. 3 shows one example of a rendering on `OMFull` as well as all the environments we tested with their relative difficulty.

In MiniGrid, all environments are of size $K \times K$ ($K$ is environment-specific) where each tile contains an object: wall, door, key, ball, chest, etc. The action space is defined as turn left, turn right, move forward, pick up an object, drop an object, and toggle an object (e.g., open or close a door). **MR** consists of a series of rooms connected by doors and the agent must open the door to get to the next room. Success is achieved when the agent reaches the goal. In **KC**, the agent has to explore the environment to find the key and open the door along the way to achieve success. **OM** is the hardest: the doors are locked, the keys are hidden in boxes, and doors are obstructed by balls.

**Results**. NovelD manages to solve all the static environments in MiniGrid. In contrast, all baselines solve only up to medium-level tasks and fail to make any progress on more difficult ones. Note that some medium-level tasks we define here are categorized as hard tasks in RIDE and AMIGo (e.g., `KCS4R3` is labeled as "KCHard" and `KCS5R3` is labeled as "KCHarder" in their works). Fig. 4 shows the results of our experiments. Half of the environments (e.g., `KCS6R3`, `OM1Q`) are extremely hard and all the baselines fail. In contrast, NovelD easily solves all listed above without any curriculum learning. We also provide the testing performance for NovelD and all baselines in Tab. 3. The results are averaged across four seeds and 32 random initialized environments.

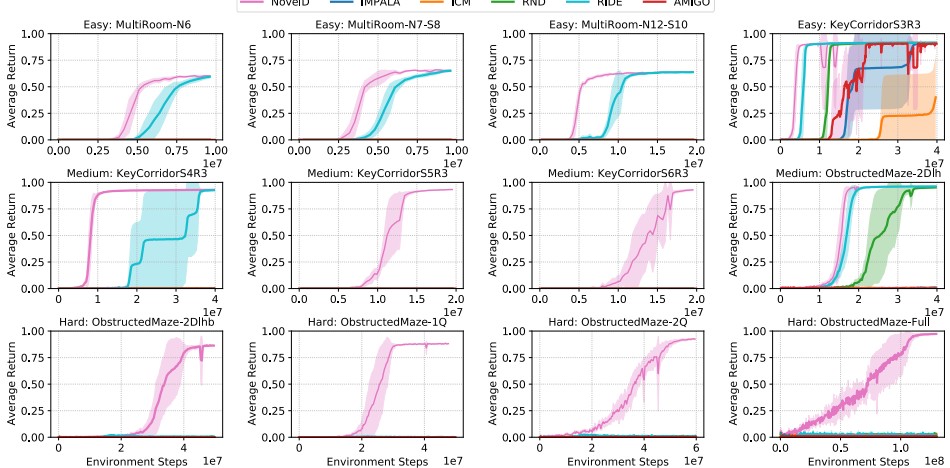

Figure 4: Results for various hard exploration environments from MiniGrid. NovelD successfully solves all the environments while all other baselines only manage to solve two to three relatively easy ones.

Table 1: Visitation counts for the toy corridor environment after 3K episodes. NovelD explores corridors more uniformly than count-based approaches.

| | C1 | C2 | C3 | C4 | Entropy |
|---|---|---|---|---|---|
| Length | 40 | 10 | 30 | 10 | – |
| Count-Based | 66K ± 28K | 8K ± 8K | 23K ± 35K | 13K ± 18K | 1.06 ± 0.39 |
| NovelD Tabular | 26K ± 2K | 28K ± 8K | 25K ± 6K | 29K ± 9K | **1.97 ± 0.02** |
| RND | 0.2K ± 0.2K | 70K ± 53K | 0.2K ± 0.07K | 26K ± 44K | 0.24 ± 0.28 |
| NovelD | 27K ± 6K | 23K ± 3K | 31K ± 12K | 26K ± 8K | 1.96 ± 0.05 |

Table 2: Entropy of the visitation counts of each room. Such state distribution of NovelD is much more uniform than RND.

| | 0.2M | 0.5M | 2.0M | 5.0M |
|---|---|---|---|---|
| Room1 | 3.48 / **3.54** | 3.41 / **3.53** | 3.51 / **3.56** | 3.49 / **3.56** |
| Room2 | **2.87** / – | 3.09 / **3.23** | 3.51 / **3.53** | 3.35 / **3.56** |
| Room3 | – / – | – / – | – / **4.02** | 3.42 / **4.01** |
| Room4 | – / – | – / – | – / **2.74** | 2.85 / **2.87** |

\* Results are presented in the order of "RND / NovelD".

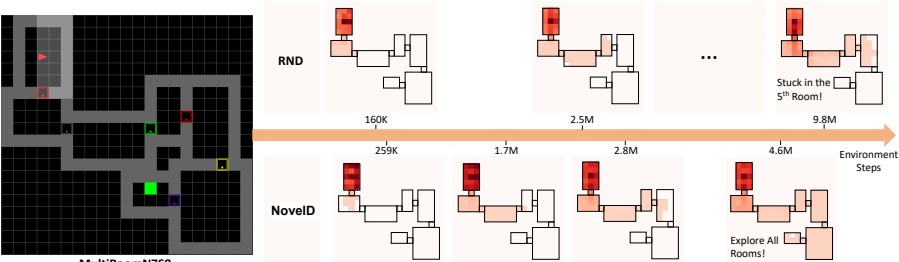

Figure 5: Normalized visitation counts $N(\mathbf{s}_t)/Z$ ($Z$ is a normalization constant) for the locations of agents. NovelD successfully explores all rooms within $4.6$M steps while RND gets stuck in the fifth room within $9.8$M steps.

Multi-Room environments are relatively easy in MiniGrid. However, all the baselines except RIDE fail. As we increase the room size and number (e.g., MRN12S10), NovelD can achieve the goal quicker than RIDE. Our method quickly solves these environments within 20M environment steps.

On Key Corridor environments, RND, AMIGo, RIDE, IMPALA and NovelD successfully solves KCS3R3 while ICM makes reasonable progress. However, when we increase the room size (e.g., KCS5R3), none of the baseline methods work. NovelD manages to solve these environments in 40M environment steps. The agent demonstrates the ability to explore the room and finds the corresponding key to open the door in a randomized, procedurally generated environment.

Obstructed Maze environments are also tricky. As shown in Fig. 4, RIDE and RND manage to solve the easiest task OM2Dlh which doesn't contain any obstructions. In contrast, NovelD not only rapidly solves OM2Dlh, but also solves four more challenging environments including OMFull. These environments have obstructions blocking the door (as shown in Fig. 3) and are much larger than OM2Dlh. Our agent learns to move the obstruction away from the door to open the door and enter the next room in these environments. This "skill" is hard to learn since no extrinsic reward is assigned to moving the obstruction. However, learning the skill is critical to achieving the goal.

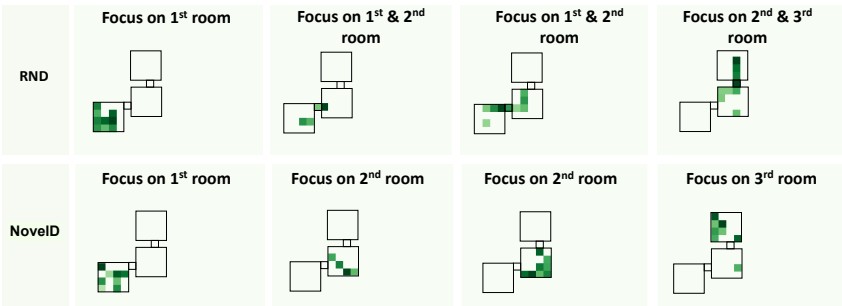

Figure 6: IR heatmaps for three-room environment. We can see that the high-IR region of NovelD has a more clear focus (the frontier) comparing to RND.

## 4.2 Analysis of Intrinsic Reward in Pure Exploration

We observe that NovelD leads to a more focused exploration at the boundary and broader state coverage by only using IR in the environment ("pure exploration" setting).

**Broader State Coverage in Long-Corridor Environment.** We design a toy environment with four disconnected corridors with lengths 40, 10, 30, and 10 respectively. The agent starts from the common entry of the corridors and picks which corridor to enter. In this example, there is no extrinsic reward, and the agent's exploration is guided by IR. We use deep Q-learning (i.e,. $Q$ function is parameterized by a deep network) and try various IRs. This includes tabular IR (count-based and NovelD tabular) and neural-network-approximated IR (RND and NovelD) respectively for this experiment. We remove clipping from NovelD for a fair comparison. Tab. 1 shows the visitation counts across 4 runs w.r.t. each corridor after 600 episodes of training. NovelD tabular explores all four corridors uniformly. On the other hand, count-based approaches focus on only two out of four corridors. NovelD also shows much more stable performance across runs as the standard deviation is much lower than RND.

**Visitation Counts Analysis in MiniGrid.** To study whether NovelD yields a broader state coverage in MiniGrid, we test NovelD and RND in a *fixed* (instead of procedurally generated for simplicity) `MRN7S8` environment. It contains 7 rooms connected by doors. We define two metrics to measure the effectiveness of an exploration strategy: (1) visitation counts $N(\mathbf{s})$ at every state over training, and (2) entropy of visitation counts *in each room*: $\mathcal{H}(\rho'(\mathbf{s}))$ where $\rho'(\mathbf{s}) = \frac{N(\mathbf{s})}{\sum_{\mathbf{s} \in \mathcal{S}_r} N(\mathbf{s})}$.

Fig. 5 shows the heatmap of normalized visitation counts $N(\mathbf{s}_t)/Z$, where $Z$ is the normalization constant. At first, RND enters room 2 faster than NovelD. However, NovelD consistently makes progress exploring new states and discovers all the rooms in 5M steps, while RND gets stuck in room 5 even trained with 10M steps.

In Tab. 2, the entropy of distribution in each room $\mathcal{H}(\rho'(\mathbf{s}))$ for NovelD is larger than that of RND. This suggests that NovelD encourages a more uniform exploration of the states than RND.

**Intrinsic Reward Heatmap Analysis in MiniGrid**. We visualize the IR produced in our algorithm and study whether it has a higher focus on the boundary. We generate the plot by running the policy from different checkpoints for 2K steps and plot the IR associated with each state in the trajectory. States that do not receive IR from the sampled trajectories are left blank. From Fig. 6, we can see that before opening the door to the 2nd room, the NovelD IR is only high in the first one. As soon as the agent opens the door, the boundary of the explored region gets pushed to the next one. This can also be illustrated in the figure that regions with high IR are also changed correspondingly. In other words, this shows that when the door between two rooms becomes a bottleneck for exploration, IR of NovelD focuses on solving this. A similar phenomenon happens between the 2nd and 3rd rooms. In contrast, the IR in RND is more spread out (e.g., both 1st and 2nd room in the second figure).

## 4.3 Ablation Study

**Episodic Restriction on IR and Clipping n NovelD**. To illustrate the importance of exploring via novelty difference, we compare NovelD with a modified version of RND: RND+ERIR. ERIR only gives RND intrinsic reward when visiting a new state for the first time in an episode. As shown in Fig. 7, the ERIR also improves the performance of RND, but NovelD has much better results. We remove ERIR from NovelD and performance also drops.

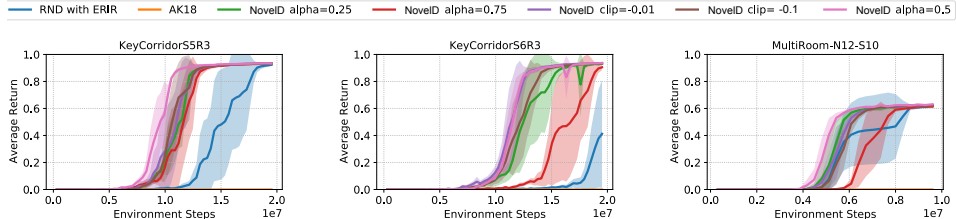

Figure 7: Ablation Study on NovelD with different scaling factor $\alpha$ and clipping threshold $\beta$. Results show that there exist an optimal choice for $\alpha = 0.5$ and $\beta = 0$.

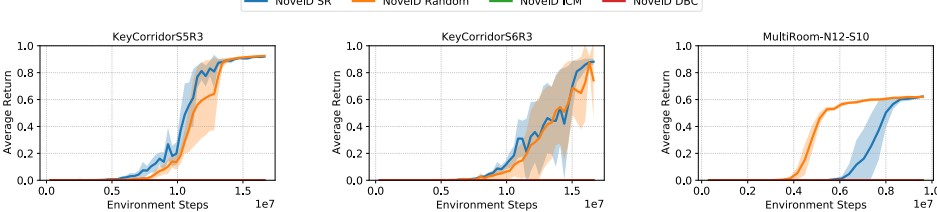

Figure 8: Ablation for NovelD under different representations. Successor representation sometimes works better than random feature, but not consistently.

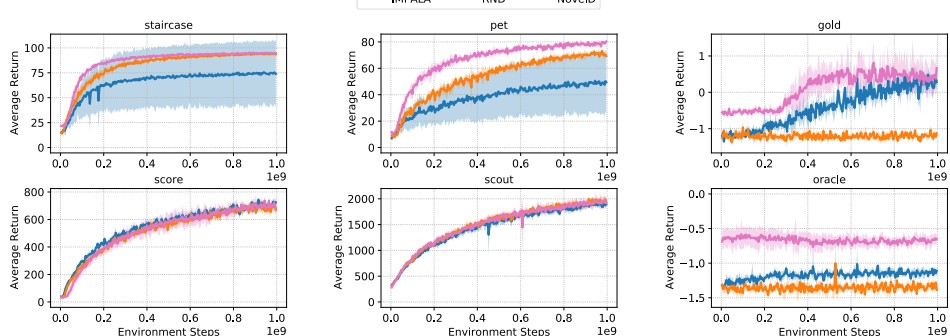

Figure 9: Results for tasks on NetHack. NovelD achieves the SOTA results comparing to RND and IMPALA.

**The optimal scaling ratio and clipping factor**. We did an ablation on how different scaling ratio $\alpha$ and clipping factor $\beta$ is affecting the performance of NovelD. Specifically, with the hyperparameter $\alpha$ and $\beta$, the intrinsic reward now takes the form:

$$r^i(\mathbf{s}_t, \mathbf{a}_t, \mathbf{s}_{t+1}) = \max\left[\text{novelty}(\mathbf{s}_{t+1}) - \alpha \cdot \text{novelty}(\mathbf{s}_t), \beta\right], \tag{4}$$

Results show that $\alpha = 0.5$ and $\beta = 0$ works the best. We also compare with AK18 [15] since they use a difference-based IR as well. AK18 achieves *zero-reward* so we don't run them on all the tasks.

**NovelD under Different Representations.** To study how NovelD performs under different representations, we set both the target $\phi(\boldsymbol{v})$ and predictor network $\phi'(\boldsymbol{v})$ in RND to be MLP and uses the same CNN-based encoders $\boldsymbol{v} = f(\mathbf{s})$ to project the image to a vector for both of them. We trains another CNN-based encoder $f'(\mathbf{s})$ according to different criteria and replace $f(\mathbf{s})$ with $f'(\mathbf{s})$ periodically. Note that once we changed the representation of states, the novelty measure changes. So we store all the samples and retrain the predictor and target from scratch. We tries four different representations of the encoder: **ICM** [49] tries to learn a compact representation that are informative about state transition, **Random** embedding, **DBC** [62] follows the criterion of aggregating model-irrelevant states and Successor Features [37] of random embedded states captures the expected future features $(f'(\mathbf{s}_t) = \mathbb{E}[\sum_{t=0} \gamma^t \phi(s_t)]$ where $\phi$ is a random embedding network).

We see that from Fig 8, DBC and ICM got zero reward since their representations are very compact and every state $s$ look similar to the RND networks. Thus, the predictor network quickly approximates the target network well and induces zero IR. For successor representation of random embedding, it works better than random in KC series of maps but worse in MR maps.

## 4.4 Atari Games

Atari games are common benchmarks for RL exploration. We test NovelD on multiple hard exploration tasks (e.g., MonteZuma's Revenge, Gravitar, Solaries). We use the same paradigm as RND [11]

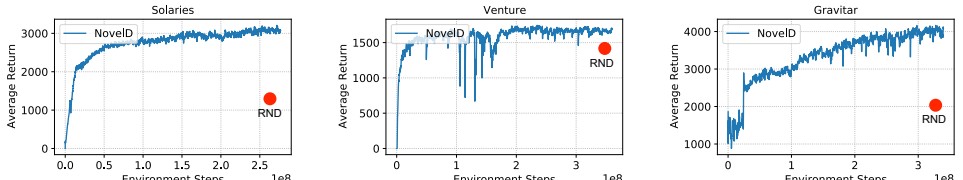

Figure 10: Results for hard exploration Atari games on Venture, Solaries and Gravitar. NovelD achieves strong results comparing to RND.

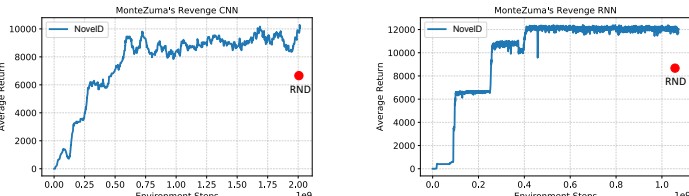

Figure 11: Results for hard exploration MonteZuma's Revenge. NovelD achieves good performance on both CNN and RNN networks comparing to RND.

and the same set of hyperparameters on 128 parallel environments. We use the code base from RND and 84x84 image of the game as input. In NovelD, same RNN network architecture is used for all the tasks (except for the fourth figure Fig. 11, we use simple CNN architecture). The baseline algorithm being used is PPO. Due to the computation limit, we didn't run NovelD to 2 billion steps as originally reported in the RND paper. We mark the performance of RND at the similar steps as red dot. Note that we didn't compare with SOTA method Go-Explore since it involves a lot of hand-tuned hyperparameters and highly incorporates domain knowledge. For another strong baseline NGU [5], since there is no open-sourced implementation online, it is hard for us to evaluate their code and do a fair comparison with them.

In Fig. 11, we can see that using CNN-based model, NovelD achieves approximately 10000 external rewards after two Billion frames while the performance reported in RND [11] is around 6700. When using an RNN-based model, NovelD reached around 13000 external rewards in 100K updates while RND only achieves 4400. In Fig. 10, we also see that NovelD manage to explore 23 rooms in the given steps while RND only explores 17. For other games (e.g., Venter, Gravitor and Solaries), NovelD is consistently better than RND in the given steps, achieving 500 to 1000 more rewards.

For the implementation of ERIR, a simple pixel-based hash table is used for counting. We found that the hash table gives reasonable performance even in the stochastic environments. This is different from Go-Explore [17] where heavy domain-knowledge is incorporated.

## 4.5 The NetHack Learning Environment

We also evaluate NovelD in NetHack Learning Environment [34], a more challenging and realistic environment. The player needs to descend over 50 procedurally-generated levels to the bottom and find "Amulet of Yendor" in the dungeon. The procedure can be described as first retrieve the amulet, then escape the dungeon and finally unlock five challenging levels (the four Elemental Planes and the Astral Plane). We test NovelD on a set of tasks with tractable subgoals in the game: **Staircase**: navigating to a staircase to the next level, **Pet**: reaching a staircase and keeping the pet alive, **Gold**: collecting gold, **Score**: maximizing the score in the game, **Scout**: scouting to explore unseen areas in the environment, and **Oracle**: finding the oracle (an in-game character at level 5-9).

Results in Fig. 9 show that NovelD surpasses PPO+RND and IMPALA (without exploration bonus) on all tasks. Especially on **Pet**, NovelD outperforms the other two with a huge margin. This again illustrates the strong performance of NovelD in an environment with huge action spaces and long-horizon reward. **Oracle** is the hardest task and no approaches are able to find the oracle and obtain a reward of 1000. NovelD still manages to find a policy with less negative rewards (i.e., a penalty of taking actions that do not lead to game advancement, like moving towards a wall). [2]

## 5  Discussion

---

[2]On NetHack, we do not report RIDE performance. Our attempt fails and the authors verify it doesn't work yet on NetHack.

**Determinstic and Stochastic Environments.** NovelD can both work with deterministic(MiniGrid) and stochastic(NetHack and Atari) environments. Although reaching the frontier of a stochastic environment is itself a hard problem, by putting a high IR on the *boundary* states and training the agent with RL algorithms, the policy will prefer to reach those frontier states more frequently. In such case, NovelD also manage to achieve its exploration purpose. In addition to that, we also test NovelD in stochastic environments like NetHack [34] and Atari games, showing superior performance. Finally, the hash-tabled based ERIR do have some potential issue in stochastic environments. However, in all of the tasks we tested, it doesn't show a clear problem of directly applying it in stochastic environments.

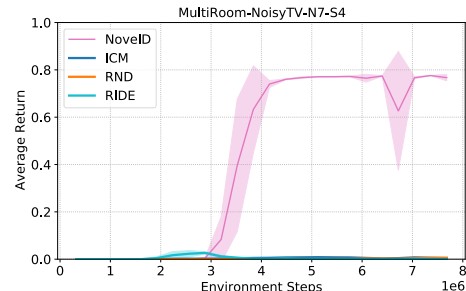

Figure 12: Results for NovelD under the Noisy-TV setting introduced in RIDE [52]. NovelD still have strong performance under the noisy TV environment.

**Noisy TV problem**. As mentioned in [11], curiosity-based and count-based criterion will get stuck in nosity-TV problem. However, RND manages to mitigate this problem [11]. Since we also adopt RND as the novelty measure, NovelD also shows the resilient to the problem. We also test NovelD in a manually-made noisty-TV setting in MiniGrid introduced in RIDE [52], where there always some blocks changing color at every time step. NovelD still gain a strong performance even under this setting. In addition, empirically we don't see the performance degrade of NovelD due to the nosiy-TV problem in all our experiments including MiniGrid, Atari Games and NetHack.

**Limitations**. We don't test NovelD in some continuous RL domains. Especially, directly applying ERIR to that for those tasks (e.g., some robotics tasks) might lead to some performance degrade. We leave a more general solution of this to future work.

# 6 Related Work

In addition to the two criteria (*count*-based and *state-diff* based) mentioned above, another stream of defining IRs is *curiosity*-based. The main idea is to encourage agents to explore areas where the prediction of the next state from the current learned dynamical model is wrong. Dynamic-AE [60] computes the distance between the predicted and the real state on the output of an autoencoder, ICM [49] learns the state representation through a forward and inverse model and EMI [32] computes the representation through maximizng mutual information $\mathcal{I}([\mathbf{s}, a]; \mathbf{s}')$ and $\mathcal{I}([\mathbf{s}, \mathbf{s}']; a)$.

Another line of research is using information gain to reward the agent. VIME [28] uses a Bayesian network to measure the uncertainty of the learned model. Later, to reduce computation, a deterministic approach has been adopted [1]. Other works also propose to use an ensemble of networks for measuring uncertainty [50, 56]. We can also reward the agent by Empowerment [33, 25, 53, 42], prioritizing the states that the agent can take control through its actions. It is different from state-diff: if $\mathbf{s}_{t+1}$ differs from $\mathbf{s}_t$ but *not* due to agent's choice of actions, then the empowerment at $\mathbf{s}_t$ is zero. Other criteria exist, e.g., diversity [20], feature control [30, 16] or the KL divergence between current distribution over states and a target distribution of states [35].

Outside of intrinsic reward, researchers have proposed to use randomized value functions to encourage exploration [46, 27, 47]. Adding noise to the network is also shown to be effective [23, 51]. There has also been effort putting to either explicitly or implicitly separate exploration and exploitation [14, 22, 36]. Go-Explore series [17, 18] also fall in this category. We might also set up different goals for exploration [26, 43, 3]. Another stream of research studies exploration problem using model-based approaches [8, 44, 45]. They either assume a world model is given or attempt to learn it, which is often impractical when the environment is complex. However, NovelD is completely model-free and can be applied widely to complicated environments.

Curriculum learning [7] has also been used to solve hard exploration tasks. The curriculum can be explicitly generated by: searching the space [54], teacher-student setting [39], increasing distance between the starting point and goal [29] or using a density model to generate a task distribution for the meta learner [21]. NovelD never explicitly generates a curriculum.

## 7 Conclusion

In this work, inspired by breadth-first search, we introduce a new criterion for intrinsic reward (IR) that encourages exploration using a regulated difference of novelty measure using RND. Based on the criterion, our algorithm NovelD successfully solves all static tasks in a procedurally generated MiniGrid environment, which is a significant improvement over the previous SOTA. In multiple environments, NovelD shows a broader state coverage with a focus of IR on the boundary states. We also evaluate NovelD on NetHack, a more challenging task. NovelD outperforms all the baselines by a large margin. In summary, this simple criterion and the ensuing algorithm demonstrate effectiveness in solving sparse reward problems in reinforcement learning, opening up new opportunities to many real-world applications.

## 8 Acknowledgements

This project occurred under the BAIR Commons at UC-Berkeley and we thanks Commons sponsors for their support. In addition to NSF CISE Expeditions Award CCF-1730628, UC Berkeley research is supported by gifts from Alibaba, Amazon Web Services, Ant Financial, CapitalOne, Ericsson, Facebook, Futurewei, Google, Intel, Microsoft, Nvidia, Scotiabank, Splunk and VMware.

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
