# A Appendix

## A.1 Final Testing Performance for MiniGrid

We provide final testing performance for NovelD and all baselines in MiniGrid. NovelD shows substantial improvement over other baselines.

Table 3: Final testing performance for NovelD and all baselines.

|  | MRN6 | MRN7S8 | MRN12S10 | KCS3R3 | KCS4R3 | KCS5R3 |
|---|---|---|---|---|---|---|
| ICM | $0.00 \pm 0.0$ | $0.00 \pm 0.0$ | $0.00 \pm 0.0$ | $0.45 \pm 0.052$ | $0.00 \pm 0.0$ | $0.00 \pm 0.0$ |
| RIDE | $\mathbf{0.65} \pm 0.005$ | $0.67 \pm 0.001$ | $0.65 \pm 0.002$ | $0.91 \pm 0.003$ | $0.93 \pm 0.002$ | $0.00 \pm 0.0$ |
| RND | $0.00 \pm 0.0$ | $0.00 \pm 0.0$ | $0.00 \pm 0.0$ | $0.91 \pm 0.003$ | $0.00 \pm 0.0$ | $0.00 \pm 0.0$ |
| IMPALA | $0.00 \pm 0.0$ | $0.00 \pm 0.0$ | $0.00 \pm 0.0$ | $0.91 \pm 0.004$ | $0.00 \pm 0.0$ | $0.00 \pm 0.0$ |
| AMIGO | $0.00 \pm 0.0$ | $0.00 \pm 0.0$ | $0.00 \pm 0.0$ | $0.89 \pm 0.005$ | $0.00 \pm 0.0$ | $0.00 \pm 0.0$ |
| NovelD | $0.64 \pm 0.003$ | $\mathbf{0.67} \pm 0.001$ | $\mathbf{0.65} \pm 0.002$ | $\mathbf{0.92} \pm 0.003$ | $\mathbf{0.93} \pm 0.003$ | $\mathbf{0.94} \pm 0.001$ |

|  | KCS6R3 | OM2Dlh | OM2Dlhb | OM1Q | OM2Q | OMFULL |
|---|---|---|---|---|---|---|
| ICM | $0.00 \pm 0.0$ | $0.00 \pm 0.0$ | $0.00 \pm 0.0$ | $0.00 \pm 0.0$ | $0.00 \pm 0.0$ | $0.00 \pm 0.0$ |
| RIDE | $0.00 \pm 0.0$ | $0.95 \pm 0.015$ | $0.00 \pm 0.0$ | $0.00 \pm 0.0$ | $0.00 \pm 0.0$ | $0.00 \pm 0.0$ |
| RND | $0.00 \pm 0.0$ | $0.95 \pm 0.0066$ | $0.00 \pm 0.0$ | $0.00 \pm 0.0$ | $0.00 \pm 0.0$ | $0.00 \pm 0.0$ |
| IMPALA | $0.00 \pm 0.0$ | $0.00 \pm 0.0$ | $0.00 \pm 0.0$ | $0.00 \pm 0.0$ | $0.00 \pm 0.0$ | $0.00 \pm 0.0$ |
| AMIGO | $0.00 \pm 0.0$ | $0.00 \pm 0.0$ | $0.00 \pm 0.0$ | $0.00 \pm 0.0$ | $0.00 \pm 0.0$ | $0.00 \pm 0.0$ |
| NovelD | $\mathbf{0.94} \pm 0.017$ | $\mathbf{0.96} \pm 0.005$ | $\mathbf{0.89} \pm 0.063$ | $\mathbf{0.88} \pm 0.067$ | $\mathbf{0.93} \pm 0.028$ | $\mathbf{0.96} \pm 0.058$ |

## A.2 Hyperparameters for MiniGrid

Following [12], we use the same hyperparameters for all the baselines. For ICM, RND, IMPALA, RIDE and NovelD, we use the learning rate $10^{-4}$, batch size 32, unroll length 100, RMSProp optimizer with $\epsilon = 0.01$ and momentum 0. We also sweep the hyperparameters for NovelD: entropy coefficient $\in \{0.0001, 0.0005, 0.001\}$ and intrinsic reward coefficient $\in \{0.01, 0.05, 0.1\}$. We list the best hyperparameters for each method below.

**NovelD**. For all the Obstructed Maze series environments, we use the entropy coefficient of 0.0005 and the intrinsic reward coefficient of 0.05. For all the other environments, we use the entropy coefficient of 0.0005 and the intrinsic reward coefficient of 0.1. For the hash table used in ERIR, we take the raw inputs and directly use that as the key for visitation counts.

**AMIGo**. As mentioned in [12], we use batch size of 8 for student agent and batch size of 150 for the teacher agent. For the learning rate, we use a learning rate of 0.001 for the student agent and a learning rate of 0.001 for the teacher agent. We use an unroll length of 100, and an entropy cost of 0.0005 for the student agent, and an entropy cost of 0.01 for the teacher agent. Lastly we use $\alpha = 0.7$ and $\beta = 0.3$ for defining IRs in AMIGo.

**RIDE**. Following [52], we use an entropy coefficient of 0.0005 and an intrinsic reward coefficient of 0.1 for the key corridor series of environments. For all other environments, we use an entropy coefficient of 0.001 and an intrinsic reward coefficient of 0.5.

**RND**. Following [12], we use an entropy coefficient of 0.0005 and an intrinsic reward coefficient of 0.1 for all the environments.

**ICM**. Following [12], we use an entropy coefficient of 0.0005 and an intrinsic reward coefficient of 0.1 for all the environments.

**IMPALA**. We use the hyperparameters introduced in the first paragraph of this section for the baseline.

## A.3 Analysis for MiniGrid

In addition to the analysis provided before, we also conduct some other analysis of NovelD in MiniGrid.

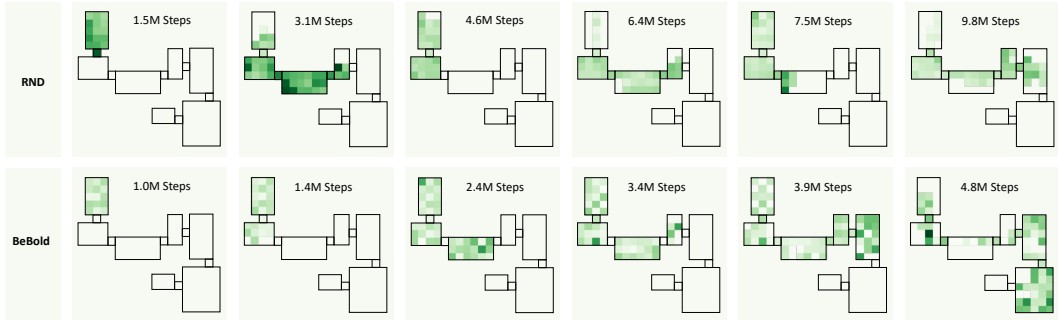

Figure 13: IR heatmaps for the location of agents. NovelD continuously pushes the high-IR area from one room to the next.

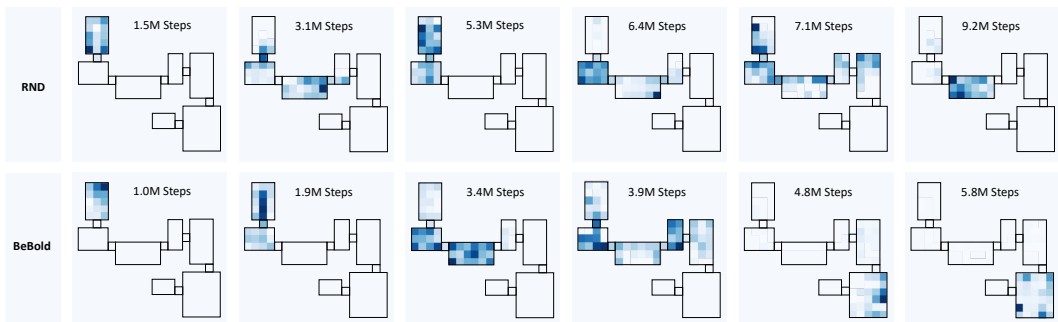

Figure 14: On policy state density heatmaps $\rho_\pi(\mathbf{s}_t)$. NovelD continuously pushes the frontier of exploration from Room1 to Room7.

**On Policy State Density in MiniGrid** We also plot the on policy state density $\rho_\pi(\mathbf{s})$ for different checkpoint of NovelD. We ran the policy for 2K steps and plot the NovelD IR based on the consecutive states in the trajectory. In Fig. 14, we can clearly see that the boundary of explored region is moving forward from Room1 to Room7. It is also worth noting that although the policy focuses on exploring one room (one major direction to the boundary.) at a time, it also put a reasonable amount of effort into visiting the previous room (other directions of to the boundary).

**More Intrinsic Analysis.** We also provide more intrinsic analysis similar to Sec. 4.2 in a seven-room environment in Fig. 13. We can see that NovelD pushes the high-intrinsic reward region to the further rooms.

### A.4 Results for All Static Environments in MiniGrid

In addition to the results shown above, we also test NovelD on all the static procedurally-generated environments in MiniGrid. There are other categories of static environment. Results for NovelD and all other baselines are shown in Fig. 15 and Fig. 16.

*Empty* (**E**) These are the simple ones in MiniGrid. The agent needs to search in the room and find the goal position. The initial position of the agent and goal can be random.

*Four Rooms* (**FR**) In the environment, the agent need to navigate in a maze composed of four rooms. The position of the agent and goal is randomized.

*Door Key* (**DK**) The agent needs to pick up the key, open the door and get to the goal. The reward is sparse in this environment.

*Red and Blue Doors* (**RBD**) In this environment, the agent is randomly placed in a room. There are one red and one blue door facing opposite directions and the agent has to open the red door then the blue door in order. The agent cannot see the door behind him so it needs to remember whether or not he has previously opened the other door in order to reliably succeed at completing the task.

*Lava Gap* (**LG**) The agent has to reach the goal (green square) at the corner of the room. It must pass through a narrow gap in a vertical strip of deadly lava. Touching the lava terminate the episode with a zero reward.

*Lava Crossing* (**LC**) The agent has to reach the goal (green square) at the corner of the room. It must pass through some narrow gap in a vertical/horizontal strip of deadly lava. Touching the lava terminate the episode with a zero reward.

*Simple CrOssing* (**SC**) The agent has to reach the goal (green square) on the other corner of the room, there are several walls placed in the environment.

### A.5 Hyperparameter for NetHack

For general hyperparameters, we use optimizer RMSProp with a learning rate of $0.0002$. No momentum is used and we use $\epsilon = 0.000001$. The entropy cost is set to $0.0001$. For RND and NovelD, we scale the forward distillation loss by a factor of $0.01$ to slow down training. We adopt the intrinsic reward coefficient of $100$. For the hash table used in ERIR, we take several related information (e.g., the position of the agent and the level the agent is in) provided by [34] and use that as the key for visitation counts.

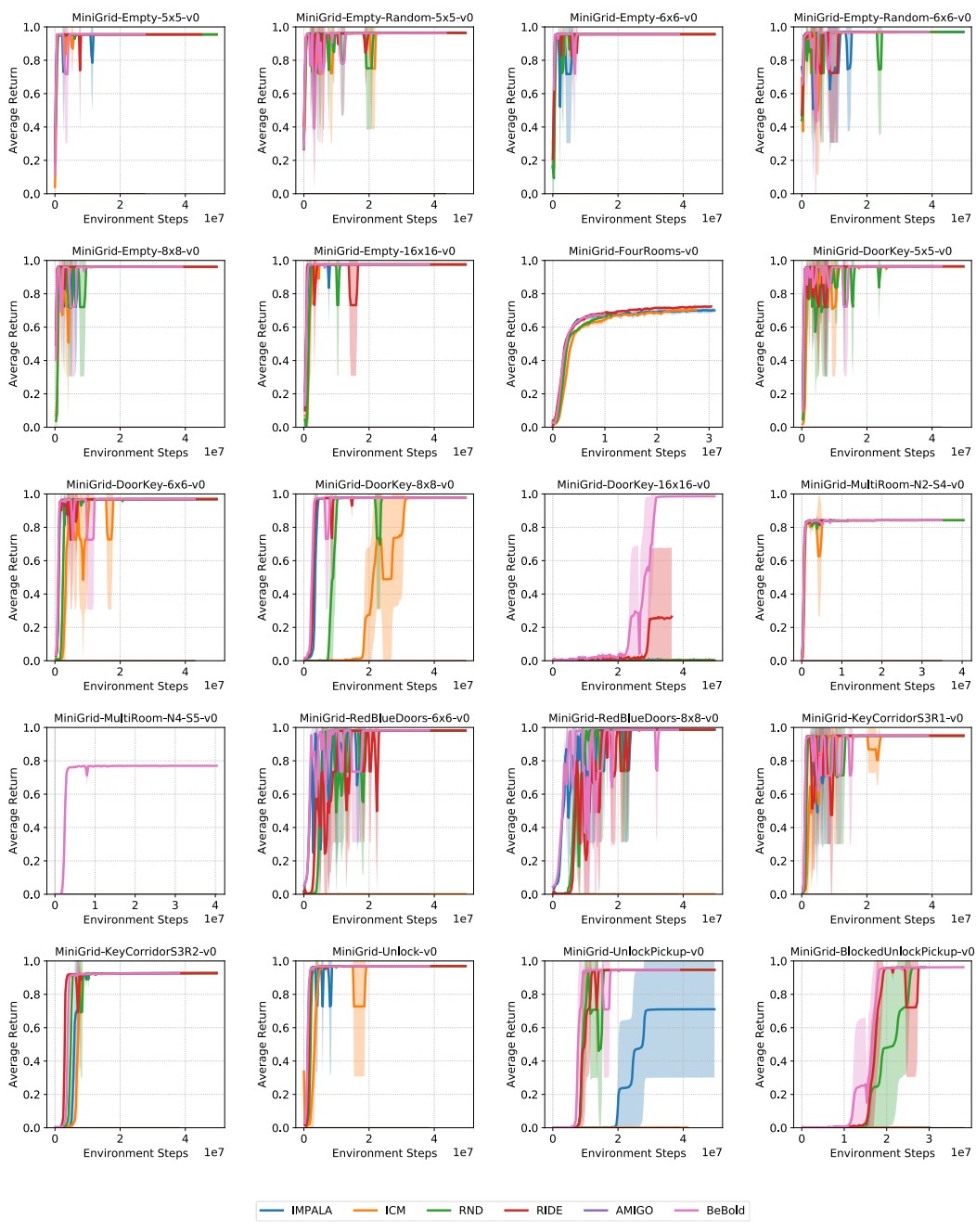

Figure 15: Results for NovelD Part 1 and all baselines on all static tasks.

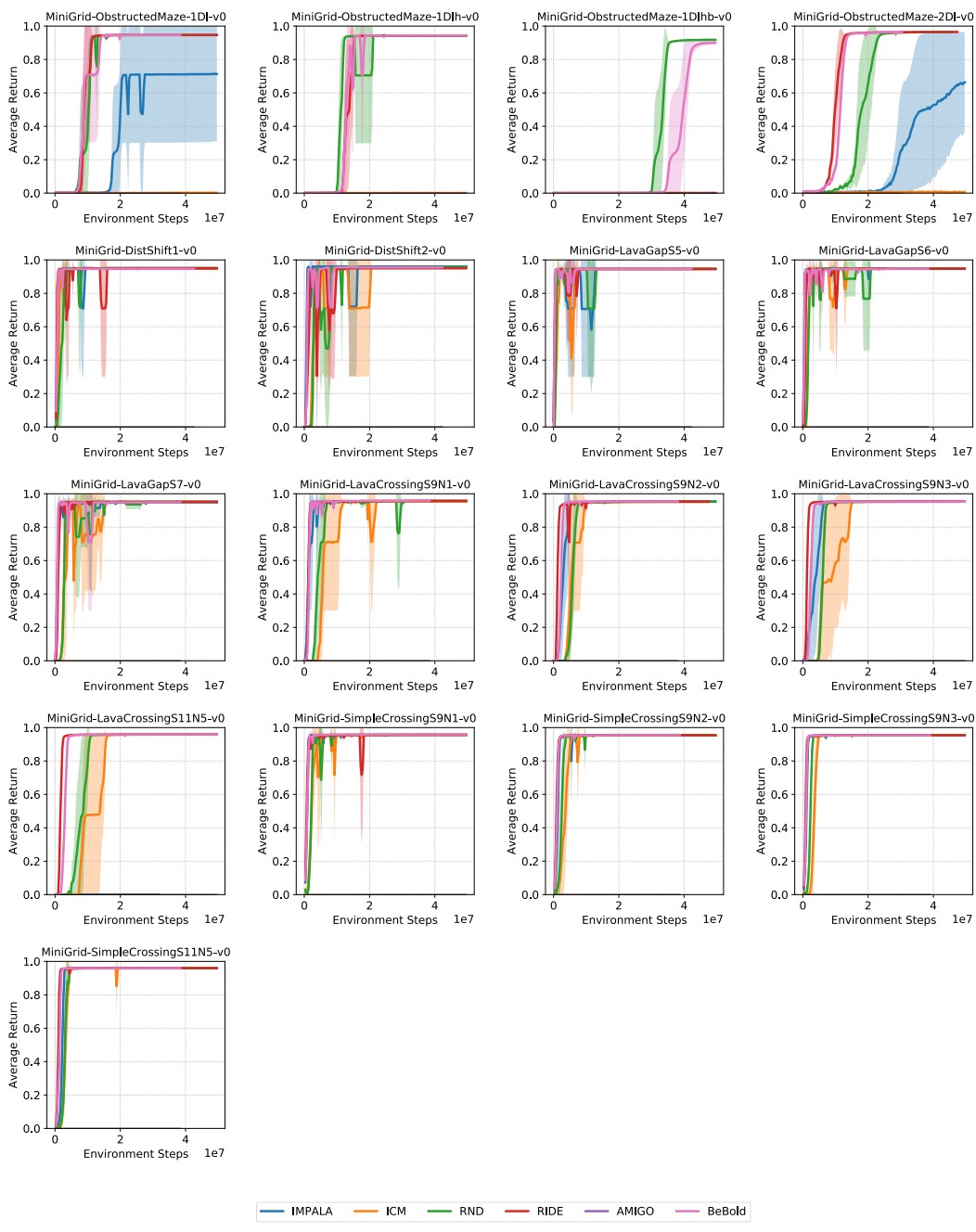

Figure 16: Results for NovelD Part 2 and all baselines on all static tasks.