# OpenReview forum: "NovelD: A Simple yet Effective Exploration Criterion"
_NeurIPS.cc/2021/Conference — NeurIPS 2021 Poster_

### Official Review · Reviewer_acG2 · 2021-07-03

**Rating:** 6
**Confidence:** 4

**Summary:**

This paper introduces a simple but effective improvement for novelty-based exploration methods: using regulated novelty difference of consecutive states in a trajectory. Extensive experiments on MiniGrid, Atari and NetHack demonstrate the effectiveness of the proposed approach.

**Limitations And Societal Impact:**

Yes

**Main Review:**

Strengths:
* The proposed modification is simple but effective, significantly outperforming prior works on MiniGrid benchmark.
* The experiments are extensive and solid.
* The writing is clear and easy to follow.

Comments:
* In Section 3.1, NovelD is described as a general meta-criterion that can be applied on top of any novelty measure. However, the authors only use RND in experiments. It would be better to test other novelty measures.
* In Section 3.2, the authors mention that one of the advantages of NovelD is asymptotic consistency. But it is unclear whether this is a severe issue. How seriously it would affect exploration if an approach is asymptotic inconsistent? The authors could add more discussions on this.
* Some implementation details are missing. The authors say they use PPO in Section 4. However, the hyper-parameters mentioned in Appendix are more like for IMPALA. Did the authors implement NovelD with PPO and other baselines with IMPALA? Is the policy a feed-forward one or recurrent one? For PPO, hyper-parameters like num_epoch, num_minibatch and GAE lambda are missing. I suggest the authors doing a through check on this and include missing implementation details.
* Some descriptions in Section 4.3 (NovelD under Different Representations) are a little bit confusing. Why "replace f(s) with f'(s) periodically"? I do not quite understand how training with different representations is done. Please give some clarifications.
* For the main results in Figure 4, it would be better to include comparisons to recent SOTAs (RAPID [1] and AGAC [2]).

Minor points (I do not factor these into my assessment):

* The authors should carefully go through the paper and fix some typos and errors:
  * Extra right parenthesis in Line 111.
  * Line 196: "Clipping n NovelD" -> "Clipping in NovelD"?
  * Line 225: Figure 11 does not have 4th figure.
* Please, use up-to-date reference instead of arXiv version, e.g., PC-PG, Agent 57, NGU. Please go over all of them.

Overall, I think the method is simple and works well as demonstrated by extensive experiments. However, the concerns need to be addressed before acceptance.

[1] Daochen Zha, Wenye Ma, Lei Yuan, Xia Hu, and Ji Liu. Rank the episodes: A simple approach for exploration in procedurally-generated environments. In International Conference on Learning Representations, 2021.

[2] Yannis Flet-Berliac, Johan Ferret, Olivier Pietquin, Philippe Preux, and Matthieu Geist. Adversarially guided actor-critic. In International Conference on Learning Representations, 2021.

**Time Spent Reviewing:**

4

---

> ### Author Response · Authors · 2021-08-10
> **Response**
>
> We thank the reviewer for the time spent reviewing our work, their thorough review, and their interest in our work. The reviewer brought up several interesting questions to which we respond below. **We have also added experiments comparing NovelD with recent SoTAs RAPID and AGAC. Please see the common questions part above.**
>
> **1. Test NovelD under different novelty measures.**
>
> We thank the reviewer for pointing this out. This is a good point. RND is one novelty measure we picked and it worked well across a variety of tasks. We'll add a simple comparison with several well-known novelty measures. However, measuring the novel of a state itself is an interesting question and very problem-specific. We’ll leave a more rigorous study of using different novelty measures under the NovelD framework to future work.
>
> **2. Is asymptotic consistency a severe problem in practice?**
>
> We agree that this might not be an issue in practice. It is actually hard to justify whether asymptotically consistency is a severe issue in practice and this is beyond the scope of this paper (we leave that to future work). However, many theoretical papers suggest this is an important property when analyzing the exploration behavior of RL agents [1, 2]. Without such property, the optimal policy will be altered by the intrinsic reward. This could be an interesting property of NovelD.
>
> **3. Implementation details for the algorithm.**
>
> We are sorry for the confusion. All the experiments (including NovelD and baselines) are implemented on top of IMPALA. We also include implementation details here and will update the paper:
> 1. We use IMPALA for NovelD and all the baselines (which propose different ways to add intrinsic rewards) on IMPALA in MiniGrid.
> 2. We use the same recurrent neural network for NovelD and all the baselines in MiniGrid.
> 3. We use a trajectory length of 100 for all the experiments in MiniGrid.
> 4. We use a batch size of 32 for all the experiments in MiniGrid.
> 5. We use an entropy cost of 5e-4 for NovelD.
> 6. We use a discounting factor of 0.99 in all the experiments in MiniGrid.
> 7. We don’t use GAE (Generalized Advantage Estimation) for all experiments in MiniGrid.
>
> **4. Some clarification on how the representation with NovelD is done.**
>
> We will clarify this part in the next revision. The reason why we want to “replace $f(s)$ with $f^\prime(s)$ periodically” is that: now we want to train the representation of a state and measure the novelty of the state under this representation by RND. Then the algorithm can be summarized as doing follows:
> 1. Initialize $f(s)$ and $f^\prime(s)$ randomly.
> 2. Train a representation of states $f^\prime(s)$.
> 3. Update $f(s)$ with $f^\prime(s)$.
> 4.  Doing exploration under the novelty measure of representation f(s).
> 5. Repeat 1-3.
>
> For step 4, the way we achieve the novelty measure under different representations is to replace the backbone of the RND network ($f(s)$) with the trained representation ($f^\prime(s)$).
>
> **5. Comparison with AGAC and RAPID.**
>
> Please see the common questions above.
>
> ----
>
> We hope that we have addressed the reviewer’s concerns. We would be happy to answer any questions and discuss further in case the reviewer believes there are any missing details.
>
> [1] Jin, Chi, et al. "Is Q-learning provably efficient?." arXiv preprint arXiv:1807.03765 (2018).
>
> [2] Cai, Qi, et al. "Provably efficient exploration in policy optimization." International Conference on Machine Learning. PMLR, 2020.

---

### Official Review · Reviewer_JUfr · 2021-07-10

**Rating:** 6
**Confidence:** 4

**Summary:**

This paper presents a simple exploration criterion NovelD. The basic idea is to use the difference of the novelties between the next state and the current state as an intrinsic reward, which is the "boundary" of the explored and unexplored regions. This idea is implemented with a max operation to avoid negative rewards, a binary indicator that only rewards the state visited for the first time, and RDN as the novelty measure. The experiments are conducted on MiniGrid, NetHack, and some hard exploration environments in Atari.

**Ethical Concerns:**

No concerns.

**Limitations And Societal Impact:**

Yes

**Main Review:**

Strengths:
1. The idea of exploration "boundary" is interesting. The intrinsic reward is simple and intuitive.
2. The results are strong. It solves all the tasks in MiniGrid in a short time. It significantly outperforms the baselines. In addition, it also shows advantages in NetHack and some hard exploration problems in Atari.
3. The paper is clearly written.

Weaknesses:
1. The paper mentions SOTA many times. However, some SOTA algorithms in MiniGrid are not discussed/compared. [AGAC](https://openreview.net/forum?id=_mQp5cr_iNy) seem to also be able to solve many tasks in MiniGrid. In particular, AGAC is shown to be able to solve KeyCorridorS8R3, which is more challenging and is not included in this paper. [RAPID](https://openreview.net/forum?id=MtEE0CktZht) can also solve many MiniGrid environments efficiently. In particular, RAPID seems to be more sample efficient than NovelD on some MultiRoom environments (such as MultiRoom-N7-S8) based on the reported results. It is unclear why these two methods are not compared in MiniGrid tasks and are not even discussed.
2. The authors noted that "NovelD puts a more aggressive restriction: the agent is only rewarded when it visits the state for the first time in an episode." It is unclear why the authors apply this aggressive restriction. How will it perform if using the scaling method in RIDE?
3. The authors noted "Asymptotic Inconsistency" is one of the advantages of NovelD. However, the authors ignore a simple strategy. One could simply anneal the coefficient of the intrinsic to zero in the training process. In this way, the policy could still maximize the extrinsic reward eventually.
4. For Atari games, the NovelD uses a simple pixel-based hash table for counting. The pixel-based hash table seems to be not well-suited for image input. A possible reason it works is that the Atari environments are not procedurally generated so that we can encounter the same states in different episodes. It is unclear how this method can be applied to procedurally generated settings, such as procgen.

Questions:
1. The authors mention "two-stage" and "one-stage" approaches. I can not find an explanation for this.
2. In procudually-generated environments, both $s_t$ and $s_{t+1}$ could be very novel if the observation space is large. MiniGrid has a very simple observation space. Can NovelD handle procedurally-generated image input, such as procgen?


**Time Spent Reviewing:**

2

---

> ### Author Response · Authors · 2021-08-10
> **Response**
>
> We thank the reviewer for the time spent reviewing our work, their thorough review, and their interest in our work. The reviewer brought up several interesting questions to which we respond below. **We have also added experiments comparing NovelD with recent SoTAs RAPID and AGAC. Please see the common questions part above.**
>
> **1. Comparison with AGAC and RAPID.**
>
> Please see the common questions above.
>
> **2. Why use a more aggressive criterion? How about putting the criterion from RIDE?**
>
> The reason why we put such a restriction is that since NovelD has a higher reward at the boundary states, without ERIR, the agent will cross the boundary back and forth in order to collect intrinsic rewards multiple times in one episode. To be more specific, if $s_{t+1}$ is novel and $s_t$ is not, the agent will go back and forth between these two states ($s_{t} \to s_{t+1}$) to collect intrinsic reward multiple times in one episode Thus, preventing this behavior is the key to the success of NovelD. We observe a little worse performance when using the scaling method in RIDE, but it still outperforms all the baselines we compared in MiniGrid.
>
> **3. Adding an annealing factor could solve the asymptotic inconsistent problem.**
>
> We agree that adding the annealing factor could resolve the issue in practice. Empirically we could put a scaling factor to the reward and anneal to zero. However, it has several issues: 1. Any intrinsic reward function (even put a reward of 1 to all state-action pairs) combined with the anneal coefficient is asymptotically consistent. However, it doesn’t mean that the design of the intrinsic reward is meaningful. The design of intrinsic reward of NovelD by itself is asymptotically consistent. It is indeed an important property in many theoretical RL papers([1, 2]) since otherwise, the intrinsic reward can alter the optimal policy. 2. How to control the rate of annealing is often tricky in practice. In practice, it is often hard to find the optimal rate of annealing. It is because the optimal annealing rate is different for different tasks.
>
> **4. The pixel-based ERIR is not well-suited for image-based tasks.**
>
> We agree that this is a valid concern. As we mentioned in the limitations, the hash-table based design is not well-suitable for the image-based inputs. We leave a more rigorous study of this to future work. However, even with the most naive design (hash-table of the raw pixels), we achieve good performance both in NetHack and Atari games. This again proves the strength and the promise of the method.
>
> **5. Please clarify the "two-stage" and "one-stage" mentioned in the paper.**
>
> We thank the reviewer for pointing this out. We’ll update the paper. By “two-stage”, we mean that the RIDE approach will first learn a representation of states (train a representation) and then use the difference of the representation as the intrinsic reward to guide exploration (train an agent). But NovelD doesn’t need the first stage, as it uses the RND approach to guide exploration which doesn’t involve a representation training stage.
>
> **5. How does the proposed method work in Procgen?**
>
> We didn’t test NovelD on Procgen since we already evaluated it on nearly 20 maps across 3 different tasks: MiniGrid (procedurally-generated environments), NetHack (complex realistic game with large-scale symbolic states), and Atari (image-based stochastic environment). We believe these experiments are extensive and sufficient to show the effectiveness of NovelD. We plan to leave Procgen as future work.
>
> ----
>
> We hope that we have addressed the reviewer’s concerns. We would be happy to answer any questions and discuss further in case the reviewer believes there are any missing details.
>
> [1] Jin, Chi, et al. "Is Q-learning provably efficient?." arXiv preprint arXiv:1807.03765 (2018).
>
> [2] Cai, Qi, et al. "Provably efficient exploration in policy optimization." International Conference on Machine Learning. PMLR, 2020.

---

### Official Review · Reviewer_a3jn · 2021-07-16

**Rating:** 7
**Confidence:** 4

**Summary:**

The paper proposes a novel exploration bonus for reinforcement learning, NovelD, which consists in adding a weighted clipped difference of the novelty of two consecutive states to the reward. The paper argues that this provides a breadth-first kind of exploration, in contrast to pure novelty based methods that would provide a more depth-first exploration. The paper provides extensive experimental validations, on a variety of environments.

**Limitations And Societal Impact:**

Yes, to the best of my understanding.

**Main Review:**

Overall, I have very few complaints about the paper. The idea provided by the paper is novel, interesting, and the presentation is clear.

My main comment is that the paper introduces two distinct contributions to exploration: the exploration bonus, based on a difference of novelty of subsequent states, and Episodic Restriction on Intrinsic Reward, which only limits intrinsic rewards to state that were only visited once in an episode. While the exploration bonus is very general and can be used in most RL problems, ERIR is arguably more ad-hoc: it requires discrete or hashable states, potentially with a well-suited hash. The authors provide an ablation of ERIR on one of the environments, and show that NovelD with ERIR outperforms RND with ERIR. However, I feel that the relative weighting of the two contributions could be better explicited, by providing:
  - Results for NovelD - ERIR on the MiniGrid environment.
  - Results for both NovelD - ERIR and RND + ERIR on Nethack and Atari.

In addition to this comment I have a few minor comments:
- There are a few english mistakes, including, but not restricted to:
  l5. in a depth-first manner
  l8. it won't suffer -> optimistic ... won't suffer
  l9-10. in a breadth-first manner
  l29. they fail
  l30. we observe that
  l32. these methods can get trapped
  l34. the detachment
  l37. approximately
  l42. compared
  l51. by a significant
  l55. is a one-stage
  l57. asymptotically consistent
  l58. are not
  l72. has not
- There is a small notational inconsistency: alpha is used twice, once for the intrinsic reward weighting, once for the weighting between the current and next state novelty.

**Time Spent Reviewing:**

4

---

> ### Author Response · Authors · 2021-08-10
> **Response**
>
> We thank the reviewer for the time spent reviewing our work, their thorough review, and their interest in our work. The reviewer brought up several interesting questions to which we respond below. **We have also added experiments comparing NovelD with recent SoTAs RAPID and AGAC. Please see the common questions part above.**
>
> **1. Some more baselines are preferred: NovelD - ERIR on MiniGrid, RND + ERIR and NovelD-ERIR on Atari/NetHack.**
>
> We thank the reviewer for the insightful suggestion. We’ll include the results in the next revision of the paper. NovelD - ERIR on MiniGrid simply achieves 0-reward, which proves the significance of ERIR. This is because, without ERIR, the agent will cross the boundary back and forth in order to collect intrinsic rewards multiple times in one episode. To be more specific, if $s_{t+1}$ is novel and $s_{t}$ is not, the agent will go back and forth between these two states ($s_{t} \to s_{t+1}$) to collect intrinsic reward multiple times in one episode. However, in stochastic environments, we found that ERIR doesn’t matter too much (in Nethack and Atari) as the same state $s_{t}$ can hardly be visited multiple times in an episode due to the stochastic nature of the environment. Thus, we didn’t run that ablation study for RND+ERIR and NovelD-ERIR on them. Please note that we have an ablation study of NovelD compared with RND+ERIR in Figure 7 for MiniGrid.
>
> In addition, note that now we only use a hash-table based ERIR on raw pixels in Atari and NetHack. Using a different approach for implementing ERIR could improve the final performance significantly. We leave this for future work.
>
> **2. The ERIR criterion seems to be ad-hoc.**
>
> We would like to emphasize that how to design the practical algorithm based on the criterion of ERIR is domain-specific, especially it can be designed differently for symbolic states and pixel-based states. We agree that there is a better design in stochastic and pixel-based games. However, the criterion itself is important for the reason we explained in the previous question. Especially, in MiniGrid, without ERIR, the agent simply archives 0-reward. In this case, we show the necessity of the criterion in certain domains.
>
> ----
>
> We hope that we have addressed the reviewer’s concerns. We would be happy to answer any questions and discuss further in case the reviewer believes there are any missing details.

---

### Official Review · Reviewer_fYfB · 2021-07-16

**Rating:** 6
**Confidence:** 4

**Summary:**

This paper introduced NoveId, a new exploration technique that can extend any exploration bonus. The new bonus is computed by looking at the difference in novelty between a new state and the previous state. The authors argue that this bonus does not suffer from detachement and improves exploration in many complex environments. These claims are substantiated by experiments on Mini Grid, the Arcade learning environment and NetHack that shows that NoveId outperforms existing exploration methods.

**Limitations And Societal Impact:**

The authors have adequately addressed the limitations and potential negative societal impact of their work

**Main Review:**

Overall I enjoyed this paper, I think that the NoveID bonus is a good idea to encourage the agent towards regions with large intrinsic reward variance. The idea is simple and builds on existing techniques nevertheless it was properly evaluated through a series  of experiments.
Exploration remains a big challenge in reinforcement learning and papers like this one making progress in this area are important.

While I agree that NoveID improves exploration it is not clear to me if it actually fixes detachement. In particular I don't agree with the explanation provided in Figure 1. I am not sure that RND would forget that it visited the upper right corner and increase the reward in 3. after decreasing it in 2. My guess would be that instead in 3 the whole right side would be light blue. In this case the agent might indeed get stuck if it is in the middle of this region however I believe the same thing would happen with NoveID. NoveID will do better here because the agent will receive a higher reward signal if it manages to reach the exploration frontier; however NoveID will not provide a better signal than RND in regions already explored.

I'm also not sure I agree with the asymptotic inconsistency mentioned in section 3.2. Once the prediction network has gathered enough data about the environment is should be able to match the embedding network, in this case I don't see why a state in representation would happen. It is also not clear why NoveID would asymptotically consistent, can the authors provide more insight here?
 I also really enjoyed the different visualizations to help understand how NoveId behaves differently from existing algorithms.

Regarding experiments I found the MiniGrid experiments in section 4.1 and 4.2 particularly interesting and they demonstrate well the benefits of NoveID. Though in section 4.3 I found it hard to draw any statistically significant conclusion from plots given that there are only 4 seeds. Given the lower computation resources needed to run MiniGrid vs other benchmarks like Atari it would have been nice to see more seeds here.

It is also hard to draw any conclusion from Atari experiments as only evaluating hard exploration games is not enough to evaluate an exploration algorithm (see [1]). Evaluating NoveID on more games is necessary to properly evaluate its performance on Atari games and see if the increased performance on hard exploration games does not lead to a decrease performance on easier exploration games. It would also be helpful to add epsilon-greedy as a baseline and mention whether or not sticky actions were used.

In the end I still lean towards acceptance given the results on MiniGrid and NetHack however the paper would stronger with a better evaluation on Atari games.

L55: "in NoveID, there is almost no hyperparameters", I don't think this is a fair statement given that NoveID adds a new hyperparameter on top of the novelty algorithm it is using.

**Clarity**

Overall I found the paper well written and easy to follow. The authors made an effort to provide many graphics to back up their claim and help the reader.


**Small typos:**

L47: Atari games is
L52: MonteZuma's Revenge instead of Montezum's Revenge
L64: What is t?
L112: Is it Eq 4 or 2? If it is 4 it is a bit confusing as this equation is only referenced many pages later.

[1] On bonus-based exploration methods in the arcade learning environment, ICLR 2020



**Time Spent Reviewing:**

7

---

> ### Author Response · Authors · 2021-08-10
> **Response**
>
> We thank the reviewer for the time spent reviewing our work, their thorough review, and their interest in our work. The reviewer brought up several interesting questions to which we respond below. **We have also added experiments comparing NovelD with recent SoTAs RAPID and AGAC. Please see the common questions part above.**
>
> **1. Does NovelD solve detachment problem and how to interpret Figure 1?**
>
> We didn’t claim that NovelD can solve the detachment problem. However, NovelD achieves a much more uniform exploration pattern empirically (please see Table 1, Table 2, and Figure 5 for reference) with a focus on the boundary states (please refer to Figure 6). We also observe that its empirical performance is much better than all other baselines.
>
> Let’s explain Figure 1 in more detail. Define **region A** as the upper right corner and **region B** as the bottom right corner. Suppose that the RND agent has visited region A before. It then heavily explores region B which is novel, causing its visitation count to be relatively higher than region A, which leads to relatively high IR in region A and pushes the agent to revisit region A again. Note that this is not due to the forgetting/detachment issue of RND. This phenomenon is also observed empirically in Appendix Figure 13 and Figure 14, where the RND agent puts a focus on previously visited rooms while the NovelD agent doesn’t exhibit that behavior.  Please also note that Figure 1 is specific to RND, other exploration methods might show different patterns.
>
> It is a valid concern that the NovelD agent might suffer from the detachment issue. However, during all of our experiments, we didn’t observe such an issue. We leave theoretical justification as the future work.
>
>
> **2. Why is NovelD asymptotic inconsistency?**
>
> Please see the common questions above. We also don't understand the sentence:
>
> "Once the prediction network has gathered enough data about the environment is should be able to match the embedding network, in this case I don't see why a state in representation would happen".
>
> Could the reviewer explain this part a little more? We are happy to answer this question and discuss it further.
>
> **3. Please run more seeds in MiniGrid.**
>
> We’ll run more seeds in the experiments of MiniGrid and add them to the next revision of the paper. However, in some hard tasks, baseline methods are not able to make any progress in any of the 4 runs. We believe such a difference is significant even if we only evaluated 4 seeds. In addition, please note that in fact, 4-5 seeds are often considered enough in many papers (e.g., RND[1], NGU[2], RIDE[3], AMIGo[4]).
>
> **4. Evaluate NovelD on easy Atari games.**
>
> We'll try to evaluate NovelD on some maps and add them to the next revision of the paper if that addresses the reviewer's concerns. However, we would like to emphasize that achieving good performance in dense-reward settings is not the focus of this paper. Like many previous works (e.g., RND[1] and NGU[2]), we focus on solving hard-exploration tasks where the reward signal is sparse and have already shown that NovelID does well on MiniGrid, NetHack and, hard exploration Atari games.
>
> Please note that we do not claim that NovelID is universally helpful for any environment. In particular, it is possible that with NovelID, the performance of environments with dense reward might stay the same (or become a little worse), while due to asymptotically consistency, the effect of NovelID will go away with a large number of samples. In such situations, the bottleneck is not due to sparse-reward / hard-exploration but is due to other factors, which is beyond the scope of our paper.
>
> **4. Adding epsilon-greedy baseline and whether the experiments use sticky actions.**
>
> We thank the reviewer for pointing this out. We'll add the epsilon-greedy baseline in the next revision. We'll also clarify that we use sticky actions in all Atari games.
>
> **5. The claim “in NoveID, there are almost no hyper-parameters” seems to be unfair.**
>
> We thank the reviewer for pointing this out. In fact, NovelD introduces two more hyper-parameters (but it is not very sensitive to them, see Figure. 7). We’ll update the paper in the next revision for this claim.
>
> ----
>
> We hope that we have addressed the reviewer’s concerns. We would be happy to answer any questions and discuss further in case the reviewer believes there are any missing details.
>
> [1] Burda, Yuri, et al. "Exploration by random network distillation." arXiv preprint arXiv:1810.12894 (2018).
>
> [2] Badia, Adrià Puigdomènech, et al. "Never give up: Learning directed exploration strategies." arXiv preprint arXiv:2002.06038 (2020).
>
> [3] Raileanu, Roberta, and Tim Rocktäschel. "RIDE: Rewarding impact-driven exploration for procedurally-generated environments." arXiv preprint arXiv:2002.12292 (2020).
>
> [4] Campero, Andres, et al. "Learning with amigo: Adversarially motivated intrinsic goals." arXiv preprint arXiv:2006.12122 (2020).

---

### Author Response · Authors · 2021-08-10
**Common Questions**

We thank the reviewers for the time spent reviewing our work, their thorough review, and their interest in our work. We'll answer some common questions here.

**1. Comparison of NovelD with AGAC and RAPID on MiniGrid.**

We summarize the results of the comparison here between AGAC, RAPID, and NovelD below and will add the comparison in the next revision of our paper. The table shows the number of steps each method needs to reach an optimal mean reward over 100 episodes for the 3 maps we tested. We ran the experiments across 4 random seeds and plotted them here. Here is the direct reading from the plot on how many environment calls are needed to learn a policy that achieves optimal reward (the smaller the better). “>4e7” here means we ran RAPID for 4e7 steps and it still didn’t reach an optimal reward.

|        | KCS4R3 | KCS5R3 | OM-1Q |
|--------|--------|--------|-------|
| AGAC   | 2.1e7  | 1.2e8  | 2.8e8 |
| RAPID  | **4.8e6**  | > 4e7  | > 4e7 |
| NovelD | 1.1e7  | **1.6e7**  | **2.3e7** |

KCS4R3 is for KeyCorridor-S4R3, KCS5R3 is for KeyCorridor-S5R3 and OM-1Q is for ObstructedMaze-1Q.

Please note that RAPID is tailored to procedurally-generated environments while NovelD is a more general algorithm that empirically shows good performance in different kinds of environments (e.g., NetHack, Atari games). It is a little unfair to compare the two algorithms in MiniGrid. We observe that RAPID seems to be more sample efficient in small maps (e.g., KC-S4R3) in MiniGrid, while for more complicated maps, NovelD performs better (e.g., OM-1Q, KC-S5R3).

In addition, NovelD is consistently better than AGAC on all three maps we tested (it shows **10x** speedup in large maps like KCS5R3 and OM-1Q). For map KeyCorridorS8R3 tested in AGAC, it is not a standard benchmark in MiniGrid and we couldn’t find the implementation of it in the AGAC code base. We have included the result for ObstructedMaze-Full in Figure 4 and NovelD is much better than AGAC: AGAC took 2e8 steps to reach a reward of 0.6, NovelD only took 1e8 steps to reach a reward of 0.95.

**2. Why RIDE is asymptotically inconsistent but NovelD isn't?**

By asymptotically consistent, we mean that when the agent collects enough data (state-action pairs), the intrinsic reward will go to 0. Please note that RND (train a predictor network to output the same embedding as a target network) doesn’t suffer from the asymptotic inconsistency issue mentioned in Section 3.2. In Section 3.2, we are referring to the bonus using difference between consecutive states in a trajectory $||\psi(s_{t+1}) - \psi(s_{t})||$ while RND uses $||\phi_{s_t} - \phi^{\prime}_{s_t}||$. The former is asymptotically inconsistent since the representation for different states is different (unless the representation of every state is the same). The latter is actually asymptotically consistent and it is what we used in our algorithm.

The reason NovelD is asymptotically consistent is, if the agent gathered enough data, then the prediction network should match the embedding of the target network. Thus, $||\phi(s_{t+1}) - \phi^{\prime}(s_{t+1})|| \to 0$ and $||\phi(s_{t}) - \phi^{\prime}(s_{t})|| \to 0$. So $||\phi(s_{t+1}) - \phi^{\prime}(s_{t+1})|| - ||\phi(s_{t}) - \phi^{\prime}(s_{t})|| \to 0$.

----

We hope that we have addressed the reviewer’s concerns. We would be happy to answer any questions and discuss further in case the reviewer believes there are any missing details.

---

### Decision · Program_Chairs · 2021-09-27

**Decision:**

Accept (Poster)

**Comment:**

After reading the reviews, the authors response and the discussions, I suggest to accept the paper. All reviewers agree that the paper presents an original idea in a clear way. Minor questions and concerns have been answered during the rebuttal by the authors. Overall the idea is simple but yet effective on a spectrum of different tasks. In addition, the method is compared with state of the art baselines. To improve the paper, experiments on more challenging environments such as 3D visual game could be considered.